

# *Cirripectes matatakaro*, a new species of combtooth blenny from the Central Pacific, illuminates the origins of the Hawaiian fish fauna

Mykle L. Hoban[1] and Jeffrey T. Williams[2]

[1] Hawai'i Institute of Marine Biology, University of Hawai'i at Mānoa, Kāne'ohe, Hawai'i, United States of America
[2] Division of Fishes, Department of Vertebrate Zoology, National Museum of Natural History, Smithsonian Institution, Washington, DC, United States of America

## ABSTRACT

Included among the currently recognized 23 species of combtooth blennies of the genus *Cirripectes* (Blenniiformes: Blenniidae) of the Indo-Pacific are the Hawaiian endemic *C. vanderbilti*, and the widespread *C. variolosus*. During the course of a phylogeographic study of these species, a third species was detected, herein described as *C. matatakaro*. The new species is distinguished primarily by the configuration of the pore structures posterior to the lateral centers of the transverse row of nuchal cirri in addition to 12 meristic characters and nine morphometric characters documented across 72 specimens and ~4.2% divergence in mtDNA cytochrome oxidase subunit I. The new species is currently known only from the Marquesas, Gambier, Pitcairns, Tuamotus, and Australs in the South Pacific, and the Northern Line Islands and possibly Johnston Atoll south of Hawai'i. Previous researchers speculated that the geographically widespread *C. variolosus* was included in an unresolved trichotomy with the Hawaiian endemic and other species based on a morphological phylogeny. Our molecular-phylogenetic analysis resolves many of the previously unresolved relationships within the genus and reveals *C. matatakaro* as the sister lineage to the Hawaiian *C. vanderbilti*. The restricted geographic distribution of *Cirripectes matatakaro* combines with its status as sister to *C. vanderbilti* to indicate a southern pathway of colonization into Hawai'i.

## INTRODUCTION

The Hawaiian Archipelago is one of the most isolated island groups in the world, constituting its own marine biogeographic province (*Briggs & Bowen, 2012*). As a result of this isolation, a high rate of endemism is found among the marine fauna of Hawai'i, with the proportion of endemism among marine fishes recorded at 25% (*Randall, 2007*) in the Main Hawaiian Islands and considerably higher on mesophotic reefs in the remote Northwest Hawaiian Islands (*Kane, Kosaki & Wagner, 2014*; *Kosaki et al., 2016*). By biomass and numeric density, endemic species may comprise 32–50% of fish assemblages on Hawaiian

Corresponding author
Mykle L. Hoban,
mh@myklehoban.com

reefs, respectively (*DeMartini & Friedlander, 2004*). Due to the volcanic origin of the archipelago, all shallow coral reef species in Hawaiʻi must by necessity originate elsewhere. Two prevailing hypotheses have been presented for the origins of Hawaiian reef species. *Hourigan & Reese (1987)* proposed that Hawaiian inshore fishes primarily originate in the western Pacific, with the closest faunal affinities found in the Ryukyu Islands and southern Japan. They suggested that rare dispersal events of Indo-Pacific species via the Kuroshio Current to the Northwest Hawaiian Islands are the most likely source of fish diversity in the archipelago. *Gosline (1955)* hypothesized that Hawaiian species originate from the south, via "stepping stone" dispersal through the Line Islands, with Johnston Atoll occupying a key position between the Hawaiian and Northern Line islands. Phylogeographic studies provide evidence that species such as limpets (*Cellana* spp.), butterflyfishes (*Chaetodon miliaris, C. fremblii*), and the deepwater snapper *Pristipomoides filamentosus* colonized from the Western Pacific (*Craig, Eble & Bowen, 2010*; *Bird et al., 2011*; *Gaither et al., 2011*) and the surgeonfishes *Acanthurus nigroris* and *A. olivaceus* and *Etelis* spp. snappers colonized via the southern route (*DiBattista et al., 2010*; *Andrews et al., 2014*; *Gaither et al., 2015*). It is apparent that there is not a single geographical origin of reef species in Hawaiʻi, but rather that taxa arrived at different times from both hypothesized origins (*Hodge, Herwerden & Bellwood, 2014*).

To date, studies examining the origins of Hawaiian reef fauna have focused primarily on conspicuous, larger-bodied species. Cryptobenthic reef fishes, as defined by *Brandl et al. (2018)*, are families with more than 10% of species smaller than 50 mm, and comprise families such as combtooth blennies (Blenniidae), gobies (Gobiidae), triplefins (Tripterygiidae), and cardinalfishes (Apogonidae), among others. Despite constituting a relatively small fraction of reef biomass, they contribute disproportionately to coral reef food webs through abundant larval supply and high predation mortality (*Brandl et al., 2019*). These fishes make up a large part of the taxonomic diversity found on reefs and can exhibit high endemism (*Brandl et al., 2018*). Hawaiʻi has 14 recognized combtooth blenny species, of which eight are endemic (*Randall, 2007*). As in other families, endemic Hawaiian blenny species often have sister taxa that are widespread in the Indo-Pacific (*Randall, 1998*).

The genus *Cirripectes* Swainson (Blenniiformes: Blenniidae) comprises 23 recognized species of combtooth blennies broadly distributed in the Indo-Pacific from East Africa to Rapa Nui (Easter Island) (*Williams, 1988*; *Delrieu-Trottin et al., 2018*). They are small (majority <100 mm SL) herbivorous and/or detritivorous fishes that primarily inhabit rocky or coral substrate in shallow (<5m) high surge forereef habitat (*Williams, 1988*). Species in the genus show considerable variation in geographic range size, from small area endemism (e.g., *C. heemstraorum*) to Indo-Pacific-wide distribution (e.g., *C. quagga*) (*Williams, 1988*; *Williams, 2010*). *Cirripectes* is most closely related to *Ophioblennius* Gill, *Scartichthys* Jordan & Evermann, and *Exallias* Jordan & Evermann (Williamsichthys clade), with which they share an ophioblennius-type larval stage having pairs of large recurved canines anteriorly on premaxillary and dentary bones (*Williams, 1988*; *Williams, 1990*; *Hundt et al., 2014*). *Cirripectes* differs from closely related genera within the clade Williamsichthys by two major characters: a transverse row of uniform-length nuchal cirri

connected by a membrane basally to form one to four groups, and a male genital structure consisting of a urogenital papilla with one or two long, tapering filaments associated with the gonopore (*Williams, 1988*).

Herein, we investigate the geographic origins of the endemic Hawaiian Scarface Blenny *Cirripectes vanderbilti* through an integrated molecular-phylogeographic-taxonomic approach. We collected specimens throughout Hawai'i and the tropical Pacific and through morphological and molecular analysis we reveal the existence of a previously undescribed species of *Cirripectes* with a limited distribution in the southern and central tropical Pacific. We describe the new species and conduct an extensive morphological examination of museum collections to ascertain its distribution. The restricted range and the phylogenetic position of the new species, combined with our molecular analyses of *C. vanderbilti* and other species in the genus, further resolve relationships within the genus *Cirripectes* and provide strong evidence for a southern route-to-colonization for the Hawaiian endemic.

## MATERIALS & METHODS

### Specimen collection

We collected specimens of the endemic Hawaiian blenny *Cirripectes vanderbilti* throughout its range in the Hawaiian Islands as well as at Johnston Atoll. We collected specimens of *C. variolosus*—the nominal sister species to *C. vanderbilti*—as broadly as possible throughout the Central Pacific. Due to the opportunistic nature of our sampling and the difficulty of capturing these species in locations where ichthyocides are prohibited, our sampling efforts were limited to Kiritimati Island, Kiribati, the Marquesas Islands, the Society Islands, and scattered locations throughout French Polynesia and the South Pacific (Fig. 1). Samples were collected by the first author and collaborators at the Northwest Hawaiian Islands and Johnston Atoll under permit #PMNM-2018-031 from the Papahānaumokuākea Marine National Monument and at Kiritimati under permit # 002/17 from the Republic of Kiribati Environment and Conservation Division. The second author and Serge Planes Centre de Recherches Insulaires et Observatoire de l'Environment, CRIOBE collected samples throughout French Polynesia under the permit "Permanent agreement, Délégation à la Recherche, French Polynesia". In addition to field collections, we expanded our geographical coverage through morphological examination of specimens of various *Cirripectes* species in the Smithsonian National Museum of Natural History in Washington, DC. Where fish were collected for this study, sampling protocols were approved by the University of Hawai'i Institutional Animal Care and Use Committee under approval number 09-753-5. This study involved no experiments on living animals.

### Procedures with specimens

We collected fishes by rotenone, pole spear, and hand-net. Specimens retained for museum collections were photographed and fixed in 10% formalin before being transferred to 70% ethanol for long-term storage. Each specimen was sub-sampled for DNA analysis before formalin fixation by removing either the right pectoral fin (in the case of museum specimens) or a portion of the caudal fin and storing it in >70% ethanol or salt-saturated DMSO solution (*Seutin, White & Boag, 1991*).
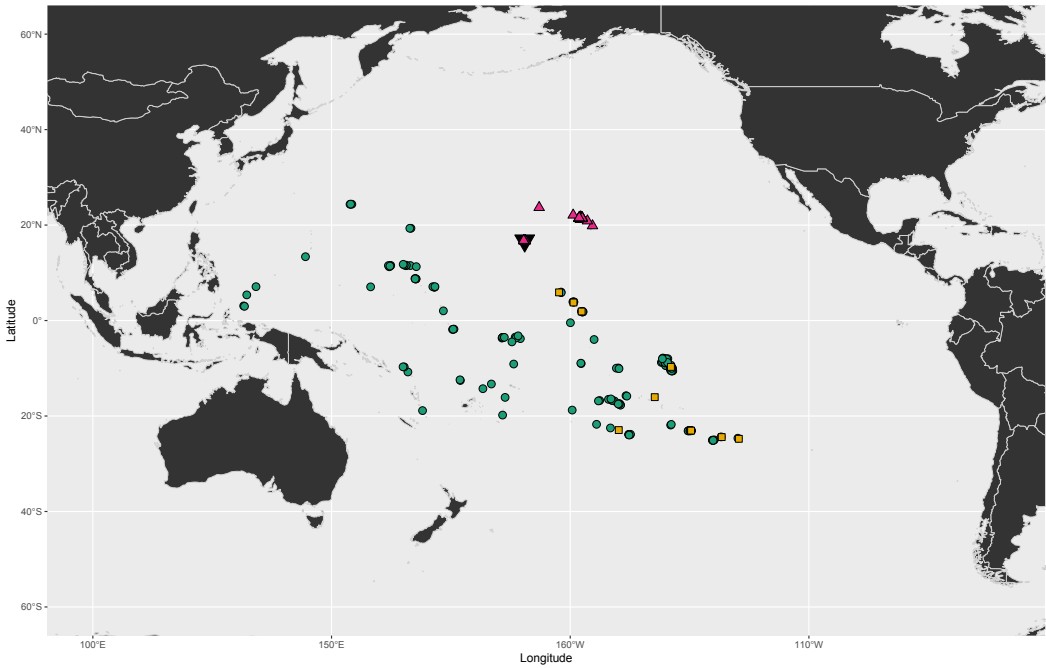

**Figure 1** **Known geographic distributions of *Cirripectes variolosus*, *C. vanderbilti*, and *C. matatakaro* sp. nov.** Green circles: *Cirripectes variolosus*. Magenta triangles: *C. vanderbilti*. Orange squares: *C. matatakaro* sp. nov. Black inverted triangle marks the location of Johnston Atoll. Occurrence data for *C. variolosus* and *C. vanderbilti* (constrained to USNM, BPBM, and CAS collections) downloaded from Global Biodiversity Information Facility (*GBIF, 2019*).

## Morphological analysis

Morphological data were taken in two categories: meristic characters and morphometrics. Detailed descriptions of counts and measures taken can be found in *Williams (1988)*, whose methods we follow and build upon. Ranges for counts are provided when variable, with holotype value given in brackets. Morphometric measurements were taken to the nearest 0.1 mm using dial calipers. Lengths of specimens are presented in mm standard length (SL), measured from the tip of the snout to the center of the caudal peduncle at the posterior edge of the hypural plate. In addition to standard meristics, we used the following characters: depth at anus (DAN) (Fig. 2A), post-orbital to mid-nuchal distance (POMN), supraorbital cirri length (SOL), left/dorsal/right nuchal separation distance (NSD/L/R), lower nuchal to opercle distance (LNO), head length (HL) (Fig. 2B), male urogenital papilla type (Fig. 3), nuchal cirri type (Figs. 4A & 4B), nuchal cirri counts (NUC), nasal cirri counts (NAC), supraorbital cirri counts (SOC), and number of lateral-line tubes (LLT). We also classified the shape and structure of the sensory pore system that lies anterior and posterior to the lateral break (where present) in the transverse row of nuchal cirri (Figs. 4B–4D). Data from morphological analyses were combined and analyzed using a principal component analysis (PCA) in R 3.6.1 (*R Core Team, 2017*). In order to reduce variation due to allometric differences, we scaled measurements proportional to SL or HL and to unit variance. The most informative principal components (PC1 and PC2) were

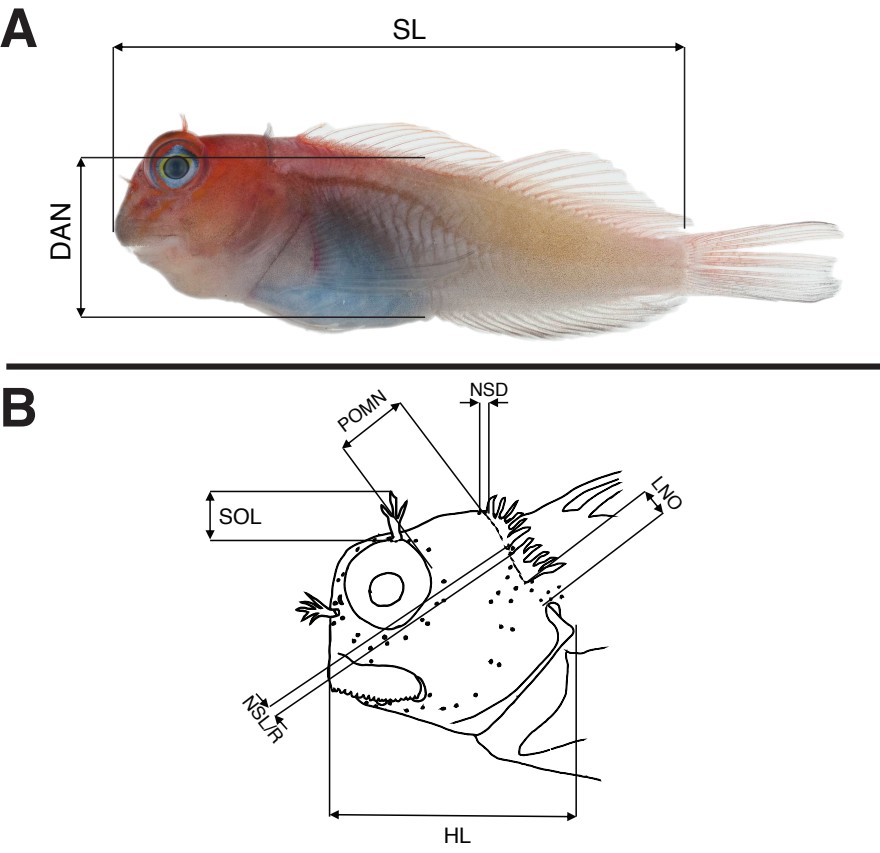

**Figure 2** **Schematic drawings showing morphological measurements for Combtooth Blennies (genus** ***Cirripectes***) **following the methods of** *Williams (1988)*. (A) Body measurements (specimen: 29 mm SL female paratype, USNM 404702, Gambier Islands). (B) Measurements of the head. For abbreviations and characters not displayed in this drawing, see Methods. Photograph in (A) by Jeffrey T Williams, Smithsonian Institution. Drawing in (B) adapted from *Williams (1986)* and used with the author's permission.

visualized in a biplot using the R package ggbiplot (*Vu, 2011*) and characters contributing most to principal component variation were identified using their loading values.

The following institutional codes are referenced to identify specimens in this study: Bernice Pauahi Bishop Museum (BPBM), National Museum of Natural History, Smithsonian Institution (USNM), California Academy of Sciences (CAS). A complete account of all specimens examined along with institutional catalog numbers is provided as supplemental information.

To better understand the geographical distribution of the new species described in this study and due to its high degree of morphological similarity to other species of *Cirripectes*, we examined BPBM and USNM fish collections to be sure that previously unidentified members of the new species were not erroneously included with other specimen lots.

## DNA extraction and sequencing

Tissue samples were sent to USNM or Hawai'i Institute of Marine Biology (HIMB) for processing. DNA from specimens sent to USNM was extracted at the National Museum

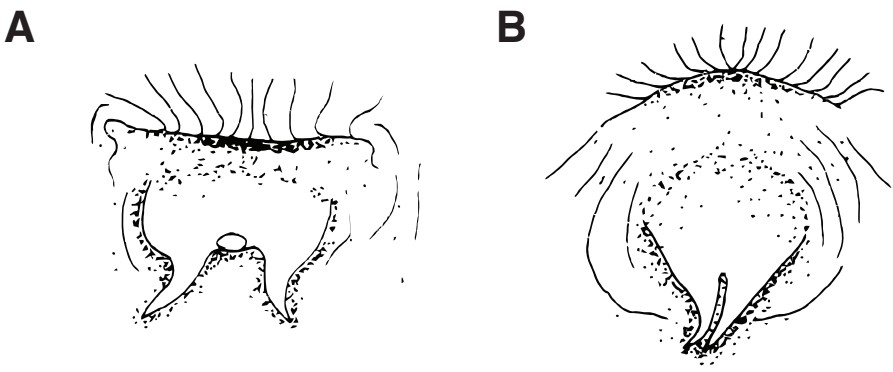

**Figure 3** **Ventral views of male urogenital papillae (anterior is up).** (A) Type I, characteristic of *Cirripectes variolosus* and *C. matatakaro* sp. nov. (B) Type II, characteristic of *C. vanderbilti.* Adapted from *Williams (1986)*, with the author's permission.

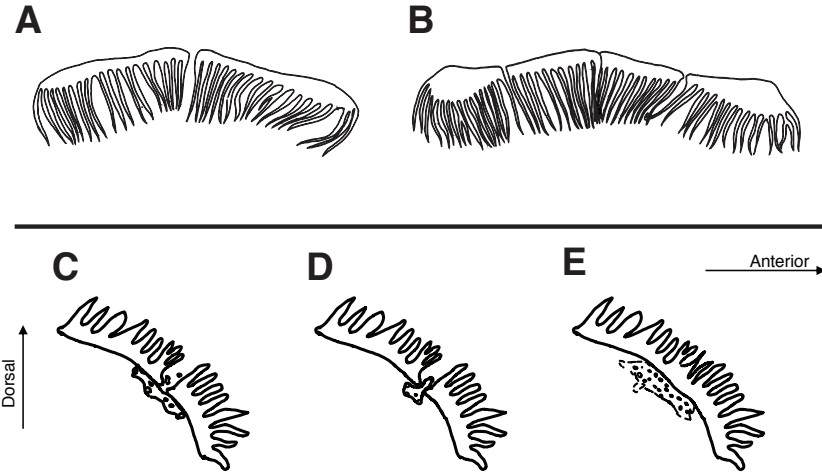

**Figure 4** **Characteristics of nuchal cirri and associated sensory pore canals.** (A) Type I nuchal cirri row: *Cirripectes vanderbilti* & *C. matatakaro.* (B) Type II nuchal cirri row: *C. variolosus.* (C–E) Structure of pore system posterior to lateral center of nuchal cirri row. (C) type I: *C. matatakaro*; (D) type II: *C. variolosus*; (E) type III: *C. vanderbilti.*

of Natural History Laboratories of Analytical Biology (LAB) with an AutoGenprep 965 (Autogen, Holliston, MA, USA) extraction robot after overnight digestion with proteinase-K in M2 buffer. Samples were amplified for cytochrome oxidase subunit I (COI) with the primers fishCOIF (TCAACYAATCAYAAAGATATYGGCAC ) and fishCOIR (ACTTCYGGGTGRCCRAARAATCA) (*Baldwin et al., 2009*) using a PCR cocktail including 5 μL GoTaq Hot Start Mix (Promega), 0.1 μL 20 μg/μL BSA, 0.3 μL each 10 mM primer and 0.5 μL dNTPs (2.5 mM each) in a total volume of 10 μL. The thermocycler profile was: 95 °C 7m, 35 cycles of 95 °C 30s, 50 °C 30s, 72 °C 45s, ending with 72 °C 2m. Samples processed at HIMB were extracted using either the EZNA-96 Tissue DNA Kit (Omega Bio-Tek, Norcross, GA, USA) following manufacturer's

protocols or a modified version of the HotSHOT protocol of *Meeker et al. (2007)* and amplified for COI using the primers FishF1 (TCAACCAACCACAAAGACATTGGCAC) and FishR1 (TAGACTTCTGGGTGGCCAAAGAATCA) following the chemistry and thermocycler settings from *Ward et al. (2005)*. DNA sequencing was performed using fluorescently-labeled dideoxy terminators on an ABI 3730XL Genetic Analyzer (Applied Biosystems, Foster City, CA, USA) at the University of Hawaiʻi Advanced Studies of Genomics, Proteomics and Bioinformatics sequencing facility and the LAB.

## Molecular and phylogenetic analyses

We aligned resolved COI sequences in Geneious 10.2.6 (https://www.geneious.com) using the CLUSTAL-W algorithm. We examined relationships within putative species groups by constructing COI haplotype networks using the R script haplonet.r (10.5281/zenodo.3532180) and a modified version of the R package pegas (10.5281/zenodo.3647668). Pairwise genetic distances were calculated using the R script gdist.r (10.5281/zenodo.3532182) and genetic summary statistics were generated with pegas (*Paradis, 2010*). Population structure was investigated using analysis of molecular variance (AMOVA) in the R packages poppr (*Kamvar, Tabima & Grünwald, 2014*) and pegas.

To assess phylogenetic relationships within *Cirripectes*, we aligned our sequences with unique COI sequences of other congeneric species from GenBank (Table 1), National Museum of Natural History Biorepository collections, and previous fish barcoding efforts in French Polynesia (*Delrieu-Trottin et al., 2019*). We used PartitionFinder 2 with greedy algorithm to select the best model(s) of nucleotide substitution partitioned by codon position, based on corrected Akaike Information Criterion (AICc) (*Guindon et al., 2010*; *Lanfear et al., 2012*; *Lanfear et al., 2016*). We rooted our reconstructions using *Plagiotremus tapeinosoma*, a confamilial outside of the Williamsichthys clade, as an outgroup. We conducted a Bayesian phylogenetic reconstruction using MrBayes 3.2.7 (*Huelsenbeck & Ronquist, 2001*; *Ronquist, 2004*), running four independent searches for 200 million generations each, saving trees every 1,000 generations and discarding the first 20% as burn-in. We verified MCMC and model parameter convergence using Tracer v1.7.1 (*Rambaut et al., 2018*). Final trees were created with R (*R Core Team, 2017*) using the ggtree package (*Yu et al., 2017*). Phylogenetic computations were run on the University of Hawaiʻi Information Technology-Cyberinfrastructure High Performance Computing cluster.

## Nomenclatural acts

The electronic version of this article in Portable Document Format (PDF) will represent a published work according to the International Commission on Zoological Nomenclature (ICZN), and hence the new names contained in the electronic version are effectively published under that Code from the electronic edition alone. This published work and the nomenclatural acts it contains have been registered in ZooBank, the online registration system for the ICZN. The ZooBank LSIDs (Life Science Identifiers) can be resolved and the associated information viewed through any standard web browser by appending the LSID to the prefix http://zoobank.org/. The LSID for this publication
**Table 1  External sequences used for phylogenetic reconstruction.**

| Species | GenBank Accession Number(s) | Sequences |
|---|---|---|
| *Cirripectes alboapicalis* | MH707846, MH707847 | 2 |
| *Cirripectes auritus* | KF489554 | 1 |
| *Cirripectes castaneus* | JQ349900, JQ349901, JQ349904, JQ349905, KX301891, MF409520, MF409642 | 7 |
| *Cirripectes chelomatus* | Published in BOLD, not entered in GenBank (BOLD BIN: BOLD:AAY8754) | 3 |
| *Cirripectes filamentosus* | KX301893 | 2 |
| *Cirripectes fuscoguttatus* | JQ431645, JQ431646, MH707851, MH707853 | 4 |
| *Cirripectes heemstraorum* | GU357568, GU357569 | 2 |
| *Cirripectes jenningsi* | MH707854, MK658251 | 2 |
| *Cirripectes obscurus* | MH707855 | 1 |
| *Cirripectes polyzona* | HQ168554 | 1 |
| *Cirripectes quagga* | KJ968000, KJ968001, KJ968004, MH707862, MH707864, MH707865 | 6 |
| *Cirripectes stigmaticus* | JQ349909, JQ349910, KF929762, KF929763, KP194481, KP194898, KP194899, MF409601 | 11 |
| *Cirripectes variolosus* | KJ968005, KU944801, MH707867–MH707872, MH707876–MH707881 | 14 |

is: urn:lsid:zoobank.org:pub:E904BAB5-9C52-46A0-9B74-530C39F571DE. The online version of this work is archived and available from the following digital repositories: PeerJ, PubMed Central and CLOCKSS.

# RESULTS

## mtDNA sequences

We resolved 641 base pairs of the mitochondrial cytochrome oxidase subunit I (COI) from 51 nominal individuals of *Cirripectes variolosus* from Kiritimati Island, Kiribati and Nuku Hiva, Marquesas, 82 individuals of *C. vanderbilti* from Hawai'i and Johnston Atoll, and five uncertain individuals initially identified as *C. variolosus* from the Marquesas, Gambier, Austral Islands, and Palmyra Atoll. New sequences generated for this study have been deposited in GenBank and are available via accession numbers MN649877– MN650012.

The Hawaiian endemic *Cirripectes vanderbilti* has relatively high haplotype diversity driven by many singleton haplotypes ($h = 0.93$), but no apparent population structure in COI across its geographic range (AMOVA, $p > 0.80$, $\Phi_{st} = -0.01$). One haplotype is shared among almost all localities (from Kure Atoll to O'ahu, including Johnston Atoll) (Fig. 5A). Due to limited sample sizes, we combined sequences from the adjacent islands Laysan, Lisianski, and French Frigate Shoals into a single population for the AMOVA analysis. The classic star-shaped pattern of the COI haplotype network for *C. vanderbilti* is indicative of a recent population expansion or high demographic turnover within Hawai'i (*Grant & Bowen, 1998*). Our collections of *Cirripectes variolosus* lacked sufficient sample sizes at most localities for population structure inference, but we detected strong differentiation between the Marquesas and Line Islands (AMOVA, $p < 0.001$, $\Phi_{st} = 0.96$).

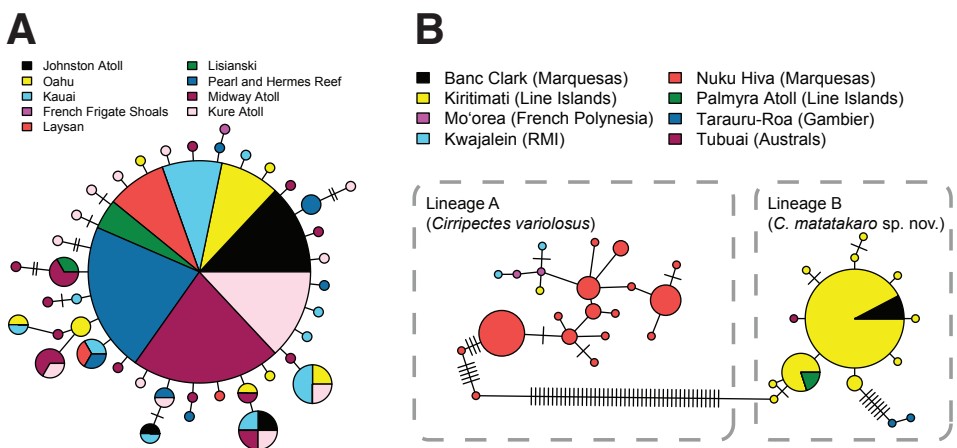

**A**

- ■ Johnston Atoll
- □ Oahu
- □ Kauai
- ■ French Frigate Shoals
- ■ Laysan
- ■ Lisianski
- ■ Pearl and Hermes Reef
- ■ Midway Atoll
- □ Kure Atoll

**B**

- ■ Banc Clark (Marquesas)
- □ Kiritimati (Line Islands)
- ■ Mo'orea (French Polynesia)
- ■ Kwajalein (RMI)
- ■ Nuku Hiva (Marquesas)
- ■ Palmyra Atoll (Line Islands)
- ■ Tarauru-Roa (Gambier)
- ■ Tubuai (Australs)

Lineage A
(*Cirripectes variolosus*)

Lineage B
(*C. matatakaro* sp. nov.)

**Figure 5** **COI haplotype networks.** Each circle represents a unique haplotype. Lines connecting circles indicate a single mutation step, with hash marks indicating additional mutations. Haplotypes are color-coded by location and circle size is proportional to haplotype frequency. Smallest circles represent haplotypes detected once. (A) *Cirripectes vanderbilti*. (B) *C. "variolosus"* (incl. *C. matatakaro* sp. nov.).

The COI haplotype networks for the specimens initially identified as *C. variolosus* (Fig. 5B) reveal two distinct lineages. One group (lineage A, *C. variolosus*) comprises the majority of individuals collected for genetic samples in the Marquesas and Society Islands, plus one from Kiritimati and the Marshall Islands. The second (lineage B, *C. matatakaro*) comprises all but one of the individuals collected from the Line Islands (Kiritimati and Palmyra) and the five "uncertain" individuals from the Marquesas, Gambier, and Austral Islands. The two haplotype lineages are separated by ∼46 mutational steps and an uncorrected pairwise distance of ∼10.8%. The Gambier Islands specimens appear to be genetically divergent within lineage B, as the haplotype network shows their COI haplotypes to be separated by a minimum of eight mutations (Fig. 5B). A BOLD search (*Ratnasingham & Hebert, 2007*) indicates that lineage A matches *C. variolosus* to 100% identity, while lineage B produces no species-level database matches with greater than 92% identity.

## Phylogenetic inference

The best-fit partitioning scheme for COI was GTR+ Γ, F81+ Γ, and GTR+I+ Γ (in order of codon position). Our COI tree (Fig. 6B) recovers two well-supported clades within the genus which correspond to nodes I and III of *Williams'* (*1988*) morphological phylogeny (Fig. 6A). Disagreements between our tree and the morphological tree may be due to cryptic lineages, lack of phylogenetic-morphological concordance, incomplete taxon sampling, and/or limitation to a single mitochondrial locus in our molecular analysis. The clade corresponding to node I contains the *Cirripectes quagga/alboapicalis/obscurus/jenningsi* species group within which specific relationships largely concur with the morphological phylogeny. The specimens identified as *C. alboapicalis* are polyphyletic, with individuals from Rapa Nui (here labeled "*C. patuki*") forming the sister group to *C. alboapicalis* and *C. obscurus* (*De Buen, 1961*; *Delrieu-Trottin et al., 2018*). *Cirripectes obscurus,* considered a Hawaiian endemic, contains two individuals from Hawai'i in addition to one from

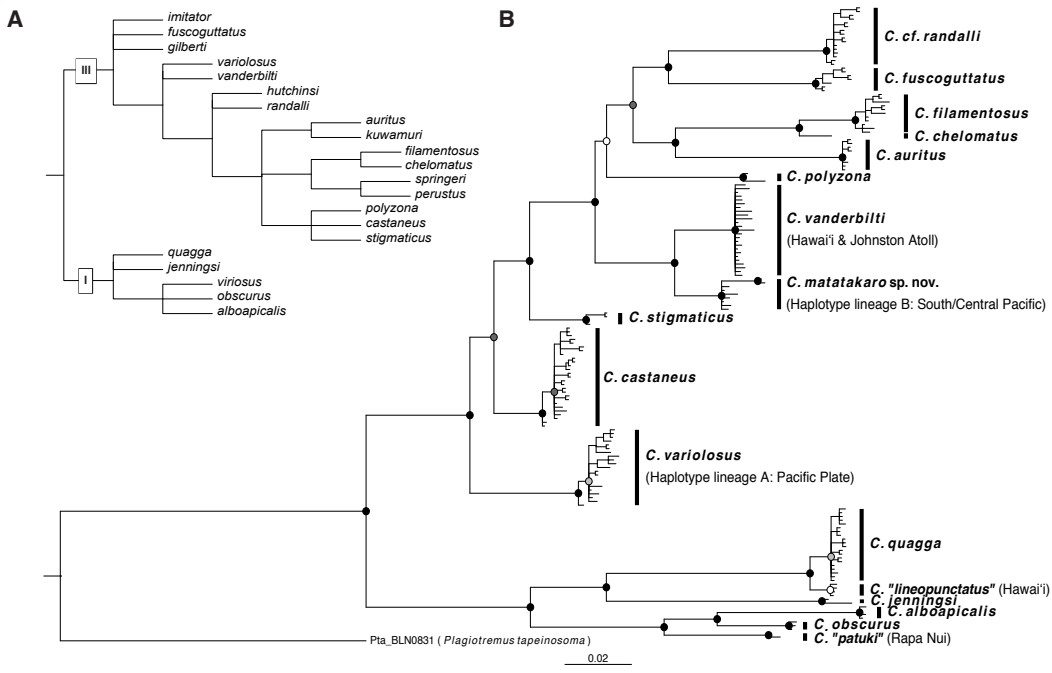

**Figure 6** **Phylogenetic hypotheses for Combtooth Blenny genus *Cirripectes*.** (A) Phylogeny based on morphological analysis, *sensu Williams (1988)*. (B) Rooted Bayesian phylogenetic reconstruction based on cytochrome oxidase subunit I (COI). Sister species *C. vanderbilti* and *C. matatakaro* sp. nov. are at right center. Node symbols are color coded by posterior probability: white: <60%, light grey: 60–75%, dark grey: 75–95%, black: >95%.

the Austral Islands (USNM 422996). Within *C. quagga,* Hawaiian individuals occur on a separate branch, here labeled "*C. lineopunctatus*" after *Strasburg (1956)*. Upon examination of the type material for *C. lineopunctatus* (USNM 164198-164201), we were unable to discern consistent differences from *C. quagga.* We did note the black-outlined white spots given in the description (*Strasburg, 1956*), although that coloration was not exclusive to the Hawaiian specimens.

The second major clade in our tree corresponds with node III of *Williams (1988)* (Fig. 6A). This group contains the new sequences (originally identified as *C. variolosus*) generated for this study and reveals that those individuals comprise two groups nested within two separate and divergent regions of the tree. Specimens from haplotype lineage A (*C. variolosus*; Fig. 5B) are shown to cluster with other individuals previously identified as *C. variolosus* in a well-supported (>0.99 posterior prob.) group basal to the other members of the clade. This position corresponds with its position in the morphological analysis. Specimens from haplotype lineage B (*C. matatakaro;* originally identified as *C. variolosus*) comprise a strongly supported (>0.99 posterior prob.) group sister to the Hawaiian endemic *C. vanderbilti*. We recover *C. filamentosus* and *C. chelomatus* as sister taxa, both closely related to *C. auritus*, which is in agreement with the morphological tree. Our tree resolves the branching order for *C. castaneus* and *C. stigmaticus*, which occur in an unresolved trichotomy in the morphological tree. *Cirripectes polyzona*, part of the same

trichotomy, is more distantly related in our analysis. However, it occurs at the only poorly supported (∼0.52 posterior prob.) internal node in our tree, so its position is not certain. A group of individuals from Réunion Island in the Indian Ocean (identified in BOLD as *C. castaneus*) cluster as the sister to *C. fuscoguttatus*. Based on the specimen photographs and their collection locality, we believe it is most likely that these are actually *C. randalli* and have tentatively labeled them as such. We lack COI sequences from *C. gilberti* or *C. hutchinsi*, however, which may also co-occur in that locality. In addition, our incomplete taxon sampling may be the cause of the longer branches between *C. randalli* and *C. fuscoguttatus*.

Our COI data resolve some of the relationships among the *Cirripectes* (particularly the previously unresolved trichotomy with *C. variolosus*, *C. vanderbilti* and other species in the genus sharing the same basic nuchal cirri morphology), and reveal the lineage labeled *Cirripectes matatakaro* as the sister lineage to the Hawaiian endemic, from which it diverges by ∼4.2% (uncorrected pairwise distance). Based on our molecular and taxonomic analyses, we conclude that *Cirripectes matatakaro* constitutes a previously undescribed species.

### *Cirripectes matatakaro* sp. nov
**Suspiria Blenny**
**urn:lsid:zoobank.org:act:B9D062E5-6D3D-4218-B225-BE31147B025B**

### Holotype
USNM 423364; adult male 60 mm SL (Fig. 7A); Tupua'i, Austral Islands, French Polynesia, south end outer reef slope with dense coral (23.4214°S, 149.44°W); depth 18–22 m; field number AUST-242; collected by rotenone and hand net; collectors JT Williams, E Delrieu-Trottin, P Sasal on 14 April 2013; vessel M/Y "Golden Shadow".

### Paratypes
USNM 409139 adult female 60 mm SL field number MARQ-139 (Fig. 7B) and USNM 409140 male 41 mm SL field number MARQ-140; Banc Clark, Marquesas Islands, French Polynesia (8.08928°S, 139.635°W); depth 17–32 m; collected by rotenone and hand net; collectors JT Williams, S Planes, E Delrieu-Trottin, P Sasal, J Mourier, M Veuille, R Galzin, T Lison de Loma, and G Mou-Tham on 28 October 2011; vessel R/V "Braveheart". USNM 404702 female 29 mm SL field number GAM-791 and UNSM 404703 male 35 mm SL field number GAM-792; Tarauru-Roa, Gambier Islands, French Polynesia, east of Mangareva, outer reef slope off Tarauru-Roa Island, corals with rock and coral rubble in the channels (23.1067°S, 134.854°W); depth 10–25 m; collected by rotenone and hand net; collectors JT Williams, S Planes, P Sasal, and E Delrieu-Trottin on 13 October 2010; vessel M/V "Claymore II". USNM 446765 female 43 mm SL field number LV12/MHCV006; Palmyra Atoll, Line Islands (5.891155°N, 162.084731°W); collected by spear and hand net; collectors M Gaither, M Iacchei, D Wagner, and D Skillings on 18 April 2010. BPBM 16928 adult male 42 mm & 43 mm SL (Fig. 8); Pitcairn Island; depth 27–30 m; collected by rotenone; collectors JE Randall, DB Cannoy, and SR Christian on 23 December 1970. BPBM 16941 female 25 mm & 53 mm SL; Pitcairn Island; depth 22–25 m; collected by rotenone; collectors JE Randall, DB Cannoy, JR Haywood, JD Bryant, and SR Christian on

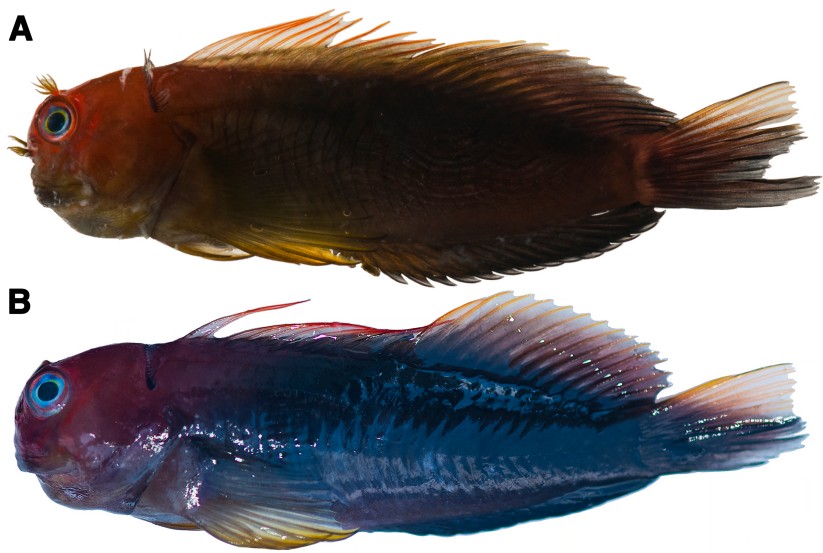

**Figure 7 Photographs of freshly dead *Cirripectes matatakaro* sp. nov. specimens showing live coloration.** (A) 60 mm SL adult male holotype (USNM 423364, Austral Islands). (B) 60 mm SL adult female paratype (USNM 409139, Marquesas Islands). Photographed by Jeffrey T. Williams, Smithsonian Institution.

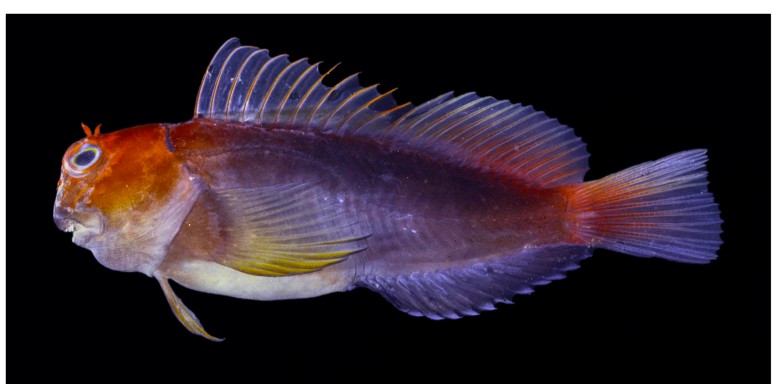

**Figure 8 Freshly dead *Cirripectes matatakaro* sp. nov. 43 mm SL adult male paratype (BPBM 16928, Pitcairn Island).** Photographed by John E. Randall (*Smithsonian Institution National Museum of Natural History, 1970*).

4 January 1971. BPBM 14060 adult female 52 mm SL; Tabuaeran Island, Kiribati; collectors EH Chave and DB Eckert in July 1972. BPBM 12270 adult female 61 mm, 54 mm, & 55 mm SL; Ducie Atoll, Pitcairn Islands; depth 30 m; collected by rotenone; collectors JE Randall, RR Costello, DB Cannoy, SR Christian, and RM McNair on 15 January 1971. CAS 48965 adult female 64 mm SL; Raroia Atoll, French Polynesia (16.0167°S, 142.4333°W); collector J Morrison for RR Harry on 8 July 1952.

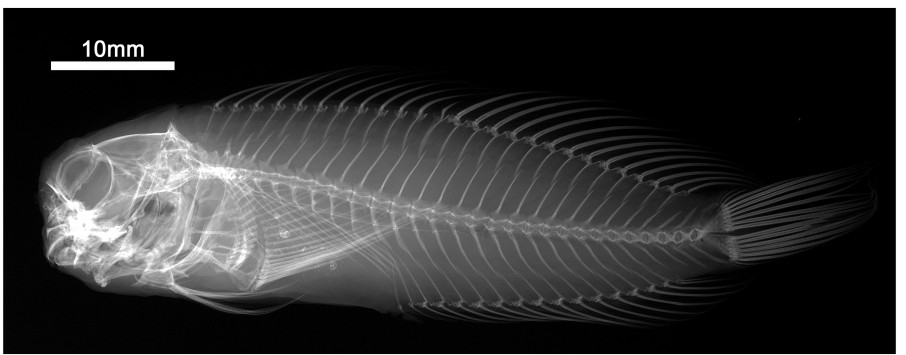

**Figure 9** **Radiograph of *Cirripectes matatakaro* sp. nov. holotype, USNM 423364, 60 mm SL, Austral Islands.**

## Diagnosis

*Cirripectes matatakaro* can be distinguished from congeners by the following combination of characters: (1) male genital papilla with two widely separated slender filaments to either side of the gonopore, type I *sensu Williams (1988)* (Fig. 3A); (2) nuchal cirri divided into two, rarely three or four, groups always slightly separated dorsally on nape with bases swollen beneath ventralmost cirri on either side; (3) overall shape of the transverse row of nuchal cirri modally type I (Fig. 4A) and sometimes type II (Fig. 4B) (types C and G *sensu* Williams), with 32–38 independently based cirri; (4) dorsal separation in row of nuchal cirri 0.1–0.7 mm (median width 0.3 mm); (5) where interrupted laterally, lateral breaks in row of nuchal cirri 0–0.4 mm in width (median width 0 mm); (6) sensory pore structure directly posterior to lateral center of row of nuchal cirri type I (Fig. 4C), posterior and parallel to row of nuchal cirri, does not visibly penetrate through break (where present); (7) 0–6 distinct LLT; (8) head coloration in life commonly bright reddish orange on upper section with bright red spots and/or slashes extending dorsally and posteriorly from the snout; (9) outer ring of iris bright orange-red in life.

## Description

Dorsal-fin rays XII,14; anal-fin rays II, 15 (anal-fin spines of sexually mature males enveloped in fleshy rugosities, females with first of two anal-fin spines embedded in swollen tissue behind gonopore–first spine visible only in radiograph or osteological preparation); total procurrent caudal-fin rays 12; caudal-fin rays 13 (nine branched); pelvic-fin rays I, 4 (spine highly reduced and difficult to discern except in osteological preparation); pectoral-fin rays 15; vertebrae $10 + 20 = 30$ (precaudal+caudal); last pleural rib on vertebral centrum 11; posteriormost anal-fin ray split through base, borne on a single pterygiophore and counted as a single ray (Fig. 9); total nuchal cirri 32–38 [34]; supraorbital cirri 6–11 [9]; nasal cirri 7–12 [10]; LLT (when present on larger individuals; tubes increase in number with increasing SL) 1–6 [4]; last LLT (when present) at position vertically below dorsal-fin ray 5–13 [12]; lower lip smooth mesially; upper lip crenulae 23–50 [49]; gill rakers 21–29 [21]; pseudobranchial filaments on left side 7–10 [9]; maxillary teeth 202–239 [uncounted]; dentary teeth 83–96 [uncounted]; nuchal cirri consisting of 32–38 [34]

independent cirri in two rows of nuchal cirri with membrane swollen ventrolaterally and rarely broken by a lateral gap halfway down; first dorsal-fin spine of adults approximately equal in length to second dorsal-fin spine (males and females); dorsal fin deeply incised above last dorsal-fin spine; dorsal-fin membrane attached to caudal peduncle anterior to caudal fin; cephalic pore system complex (six or more pores at most positions; number of pores increase with increasing SL); midsnout pores present; extra interorbital pore position with pores present; multiple pore positions behind row of nuchal cirri, pore/tube structure lies parallel to row of nuchal cirri and does not penetrate lateral break (where present) in row of cirri; male genital papilla with urogenital orifice located basally between two widely separated slender filaments (<1.0 mm in length) on a fleshy swelling behind anus (type I *sensu* Williams, Fig. 3A); maximum observed SL about 65 mm. Morphological data for selected characters of type specimens are provided in Table 2.

### Color in alcohol
Subadults and adults with cream to brown colored head and body (head is lighter in color); anterior half of the head with white slashes (0.5–1.0 mm in width) extending dorsally and posteriorly from the snout; dorsal fin translucent.

### Color in life
Adults with dark brown (though rarely pale brown to white) body; although male and female life colors may sometimes be quite different, the color pattern is highly variable and we found no consistently observed male–female color differences; dark red to orange slashes on head extending dorsal and posterior from the snout and encircling the eye; nasal and supraorbital cirri bright reddish orange; dorsal half of head bright reddish orange; nuchal cirri dark purple/brown to black; pectoral-fin color pale brown to yellow-orange; spinous portion of dorsal fin with reddish spines; dorsal-fin membrane with a translucent triangular section below the first 8–10 spines, otherwise brown below with red streaks basally; rayed portion of dorsal fin with yellowish-brown rays; upper caudal-fin rays yellowish; lower rays dark brown; anal fin dark brown; iris color silver with yellow ring around pupil and bright reddish orange ring around outer portion of eye (Figs. 7, 8 and 10).

### Comparison
*Cirripectes matatakaro* co-occurs with *C. variolosus* (its most morphologically similar congener) throughout its geographic range, but there is depth segregation in the Marquesas, Australs, Pitcairns, and Tuamotus, with *C. variolosus* typically found on reef crests at depths shallower than 5 m and *C. matatakaro* found on outer reef slopes between 10–32 m (usually greater than 20 m). Morphologically, the two differ primarily in the structure of the sensory pore canals directly posterior to the central-lateral portion of the row of nuchal cirri. *Cirripectes variolosus* has type II (Fig. 4D) pore structures whereas *C. matatakaro* has type I (Fig. 4C). *Cirripectes matatakaro* can also be distinguished in life from *C. variolosus* by the color of the outer iris ring: in *C. variolosus* it is silver-grey whereas in *C. matatakaro* it is bright reddish orange. The new species differs from its closest phylogenetic relative, *Cirripectes vanderbilti*, in the shape of the male genital papilla (*C. vanderbilti* has type II, Fig. 3B), nuchal pore structure (*C. vanderbilti* has type III, Fig. 4E), and shape of the row

Hoban and Williams (2020), *PeerJ*, DOI 10.7717/peerj.8852

**Table 2  Morphometric and meristic data for selected characters of type specimens of *Cirripectes matatakaro* sp. nov.**

| Morphometrics Sex | Holotype USNM 423364 Male | Paratypes USNM 409140 Male | USNM 409139 Female | USNM 404702 Female | BPBM 16928 (1) Male | BPBM 16941 (1) Female | CAS 48965 Female |
|---|---|---|---|---|---|---|---|
| Standard length | 57.8 | 41 | 60 | 29.3 | 42.5 | 53 | 64 |
| Head length | 16.8 | 12.8 | 15.3 | 9.6 | 12.6 | 13.9 | 16.4 |
| Depth at anus | 16 | 11.1 | 16.6 | 4.7 | 11.8 | 12.7 | 16.4 |
| Postorbital-midnuchal distance | 7 | 5 | 7.5 | 2.7 | 5.8 | 6.7 | 7.4 |
| Supraorbital cirri length | 2.2 | 0.9 | 1.8 | 0.3 | 1.7 | 1.8 | 2.1 |
| Nuchal separation, dorsal | 0.4 | 0.3 | 0.1 | 0.3 | 0.3 | 0.3 | 0.2 |
| Nuchal separation, left | 0 | 0 | 0 | 0 | 0 | 0 | 0 |
| Nuchal separation, right | 0 | 0 | 0 | 0 | 0 | 0 | 0 |
| Lower nuchal to opercle distance | 1.5 | 1.1 | 1.6 | 0.9 | 1.6 | 1.8 | 1.9 |
| Length of dorsal spine I | 8.1 | 6.9 | 10.6 | 4.7 | 11.5 | 12.2 | 13.5 |
| Length of dorsal spine II | 8.6 | 4.9 | 10.9 | 5.2 | 10.7 | 11.1 | 14.5 |
| Length of dorsal spine III | 9 | 5.3 | 10.9 | 4.9 | 10.5 | 10.5 | 12.8 |
| Length of first dorsal ray | 8.2 | 7.6 | 10 | 5.2 | 9.4 | 11.1 | 11.4 |
| Height of dorsal notch | 1.4 | 1.3 | 2 | 0.4 | 2.7 | 3.6 | 4.2 |
| **Meristics** | | | | | | | |
| Dorsal | XII, 14 | XII, 14 | XII, 14 | XII, 14 | XII, 14 | XII, 14 | XII, 14 |
| Anal | II, 15 | II, 15 | II, 15 | II, 15 | II, 15 | II, 15 | II, 15 |
| Pectoral | 15 | 15 | 15 | 15 | 15 | 15 | 15 |
| Pelvic | I, 4 | I, 4 | I, 4 | I, 4 | I, 4 | I, 4 | I, 4 |
| Nuchal cirri | 34 | 38 | 37 | 32 | 32 | 38 | 35 |
| Nasal cirri | 5 + 5 | 4 + 4 | 4 + 4 | 4 + 5 | 5 + 5 | 7 + 5 | 6 + 6 |
| Supraorbital cirri | 4 + 5 | 4 + 4 | 4 + 4 | 5 + 4 | 5 + 5 | 4 + 5 | 6 + 5 |
| Lateral line tubes | 4 | 3 | 4 | 2 | 2 | 1 | 2 |
| Last tube under dorsal spine | 12 | 10 | 12 | 6 | 10 | 9 | 10 |
| Pseudobranchial filaments | 9 + 9 | n/a | 7 + 7 | n/a | 8 + 8 | 9 + 8 | 8 |
| Gill rakers | 21 | n/a | 22 | n/a | 28 | 27 | 28 |
| Upper lip crenulations | 49 | 41 | 45 | 23 | 47 | 45 | 46 |

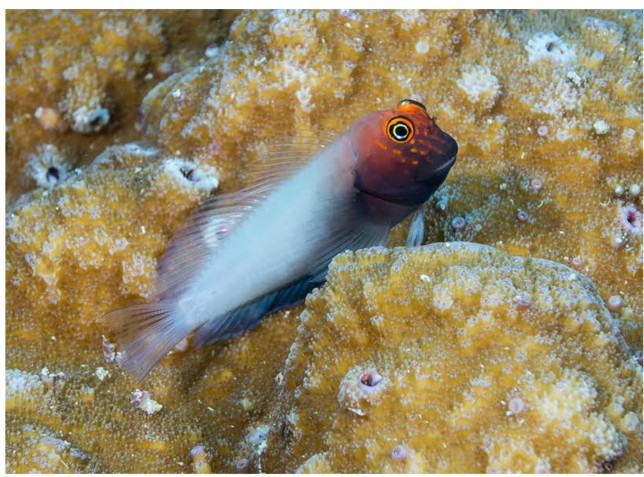

**Figure 10** Underwater photograph of likely female *Cirripectes matatakaro* sp. nov. Photographed by D. Rolla at Kiritimati Island, Kiribati, Line Islands.

of nuchal cirri itself (*C. vanderbilti* has type I, Fig. 4A). In the principal component analysis (PCA) of combined meristic and morphometric characters, PC1 and PC2 explained 27.1% and 13.6% of the total variance, respectively (Fig. 11). According to the PCA loadings, PC1 was most influenced by the ratio between the lengths of the first and last dorsal-fin spines, the HL:SL ratio, the number of nuchal cirri, the number of distinct bases on the row of nuchal cirri, and width of lateral breaks in the row of nuchal cirri (0.30, 0.25, 0.24, −0.35, −0.32 respectively). On the PC2 axis, the most influential factors were HL:SL ratio and the ratio of supraorbital cirri length to HL (0.24, −0.40). This fits the qualitative observations that *C. matatakaro* generally has greater HL relative to SL than *C. vanderbilti* or *C. variolosus* and more numerous nuchal cirri and narrower-to-absent separation at the lateral breaks in the row of nuchal cirri relative to *C. variolosus* (*C. vanderbilti* lacks these breaks entirely). The PCA biplot was clearly partitioned, indicating morphological divergence among species, particularly showing the distinct morphospace occupied by *C. matatakaro*. The two *C. matatakaro* data points outside the black normal ellipse represent the specimens from Tarauru-Roa in the Gambier Islands, indicating morphological as well as genetic differentiation in this remote locality.

## Distribution

*Cirripectes matatakaro* is known from the Northern Line Islands (Kiritimati and Palmyra), the Marquesas, and the Tuamotus and from Ducie Atoll and Pitcairn west to the Gambier and Austral Islands (Fig. 1). *Cirripectes variolosus* and *C. vanderbilti* co-occur at Johnston Atoll and to date, no specimens of *C. matatakaro* have been collected there. However, an examination of the 13 *C. variolosus* specimens from that locality housed in the USNM fish collections (USNM 198731) revealed several individuals with characters resembling the new species. In addition, several of those Johnston Island individuals grouped closely with *C. matatakaro* in the PCA biplot (Fig. 11). Those specimens were collected in 1964 and the delicate nuchal pore structures were degraded and very light in color, leaving them

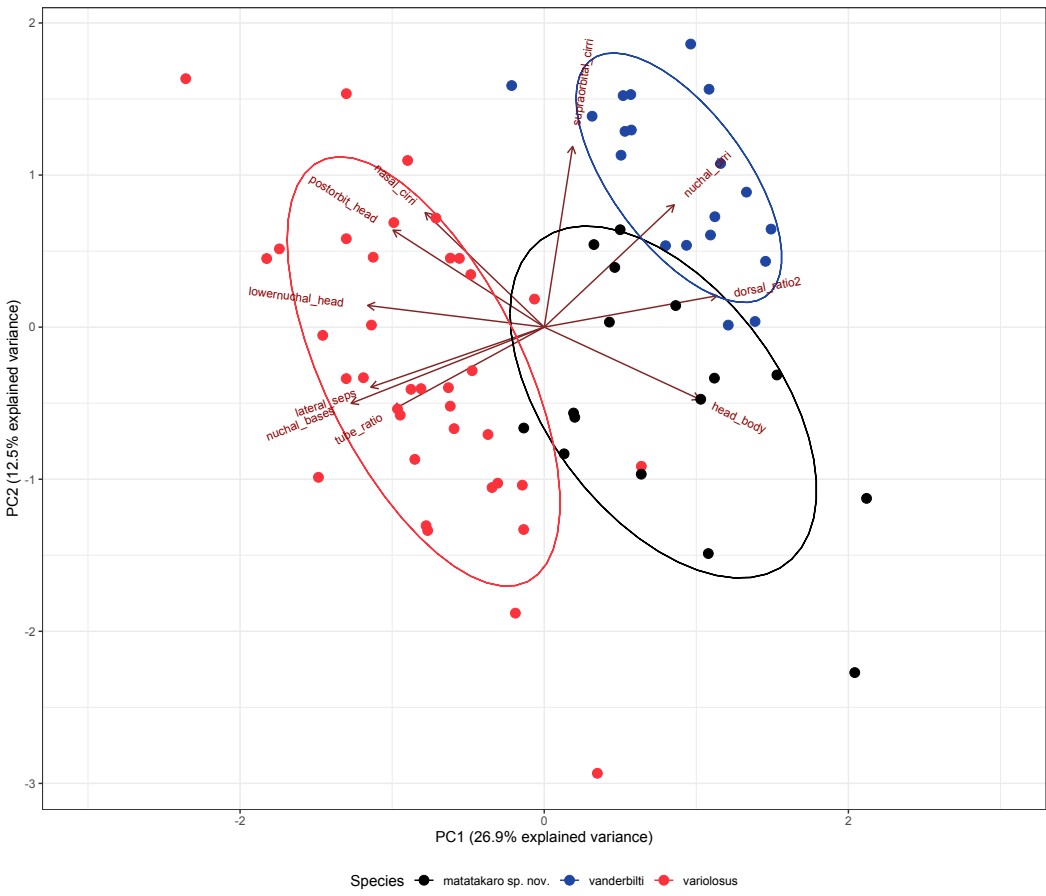

**Figure 11  Biplot of the first two principal components from a PCA of 12 meristic characters and 9 morphometric characters of 72 specimens of three Combtooth Blenny species.** Some characters combined or averaged when doubled, characters with zero variance dropped. Maroon axes indicate loading amplitudes and directions. Points represent individual specimens, color coded by nominal species: red: *Cirripectes variolosus*, blue: *C. vanderbilti*, black: *C. matatakaro* sp. nov.

difficult to identify definitively. However, we believe it is highly likely, especially given its status as sister to the Hawaiian endemic, that the range of *Cirripectes matatakaro* includes Johnston Atoll. *Williams (1988)* quoted Gosline as stating that no Hawaiian endemics occur sympatrically with their Pacific-wide counterparts at Johnston Island. The emergence of *Cirripectes matatakaro* rather than *C. variolosus* as sister to *C. vanderbilti* makes this once again an open question. In addition to the above, we conducted a thorough morphological examination of the extensive holdings of the USNM *C. variolosus* specimens from localities throughout the Pacific. Aside from the Johnston Atoll specimens already mentioned, we found no additional individuals with the characteristics of the new species. As such, we believe it is probable that the Central/Southeastern-Central Pacific distribution we have so far observed is the entire geographical range of *C. matatakaro*.

## Etymology

The specific epithet is i-Kiribati, consisting of the words "mata" (eye) and "takaro" (ember/burning coal) and refers to the large, eager-seeming eyes and the reminiscence of the red slashes on the face to smoldering embers or burning coals. The species was named in the i-Kiribati language to honor the people and culture of Kiribati, where the first author first encountered and collected the new species. The common name Suspiria Blenny is in reference to the color palette of the 1977 Dario Argento film of the same name.

## Remarks

*Cirripectes matatakaro* is noteworthy in habitat use for a member of this genus. In the southern portion of its range, from Pitcairn to the Austral Islands, *C. matatakaro* has primarily been collected from >20m depth, deeper than other known *Cirripectes* species. This may be a unique habitat exploited by this species, although in the Line Islands (Kiritimati and Palmyra) we collected it from the shallow (<5 m) oceanic forereef habitat more typically inhabited by congeners. *Williams (1988)* remarked that certain "problematical" specimens of *C. variolosus*, which were collected from deeper reefs in the Pitcairn Islands exhibited a reddish-orange head, and he speculated that the coloration might be an artifact of their depth of occurrence. Upon reexamination of that material, we determined those specimens to be the new species. Thus, the reddish color is more likely to be a property of the species rather than a product of its habitat, as individuals from Kiritimati and Palmyra show similar color patterns. We have not seen consistent evidence of sexual dichromatism, however one individual photographed *in situ* at Kiritimati Island had a distinctively light grey body coloration that is sometimes seen in females of other species of *Cirripectes* (Fig. 10).

## DISCUSSION

Geographic origins have been investigated for many conspicuous, larger-bodied Hawaiian reef fishes, but few studies have focused on endemic cryptobenthic species. Cryptobenthic species are key and often-overlooked members of reef communities despite their diminutive size and relatively low standing biomass. They contribute disproportionately to coral reef food webs through steady larval supply and high mortality by predation (Brandl et al., 2018; 2019). Combtooth blennies represent typical cryptobenthic fishes and pose an interesting phylogeographic question; they are highly sedentary as adults but may be quite dispersive as larvae; the closely related genus *Ophioblennius* showed low population structure and lacked genealogical concordance throughout the East Pacific (*Muss et al., 2001*). The species of the genus *Cirripectes* show considerable variation in range size, from small-range endemics such as *C. heemstraorum, C. randalli,* and *C. vanderbilti*, to species with very broad distributions such as *C. quagga* and *C. stigmaticus* (*Williams, 1988*; *Williams, 2010*). However, upon closer examination species with large geographic ranges have sometimes been shown to comprise multiple cryptic lineages, indicating that these fishes are less dispersive than was previously thought (*Williams, 2010*; *Delrieu-Trottin et al., 2018*). *Cirripectes variolosus,* which is known from across the Pacific Plate, has been associated in an unresolved phylogenetic trichotomy with the Hawaiian endemic *C. vanderbilti* (*Williams, 1988*). Our mtDNA phylogeny reveals

that *C. variolosus* is among the species that mask cryptic diversity. Through an integrated taxonomic approach, we determined that a subset of specimens initially identified as *C. variolosus* constitute a new species, *C. matatakaro*. The new species is morphologically similar to *C. variolosus*, but our phylogenetic reconstruction shows *C. matatakaro* to be the sister species to *C. vanderbilti*. Our tree additionally provides support for *C. patuki*, a Rapa Nui endemic, and the possibility of an antitropical distribution for *C. obscurus* (*Delrieu-Trottin et al., 2018*). The data also suggest that Hawaiian *C. quagga* individuals may comprise a distinct lineage, warranting re-evaluation and possible resurrection of *C. lineopunctatus* (*Strasburg, 1956*). The COI tree, despite some differences in internal branching order, is largely in concordance with the original 1988 morphological phylogeny (Fig. 6A & 6B). Greater taxon coverage and additional genetic markers may help resolve remaining differences as well as potentially reveal more cryptic lineages within widely distributed species. Our sampling coverage of the new species is limited by sample size and geographic distribution, but its phylogenetic position, combined with morphological analysis of museum specimens and consideration of their geographic distributions, enables us to make inferences about the Pacific origin of *C. vanderbilti*.

*Cirripectes matatakaro* has been collected from the Line Islands, Marquesas, Tuamotus, Pitcairn Islands, Gambier Islands, and Austral Islands. *Cirripectes variolosus* has been collected extensively across the Pacific Plate, from Palau to the Marquesas and throughout Micronesia, the South Pacific, and the Line Islands (Fig. 1). As the two species are morphologically very similar to one another, we sought to determine whether museum lots of *C. variolosus* might contain misidentified individuals of the new species. None of the specimens we examined from any available locality matched characters of *C. matatakaro*, with the possible exception of the two individuals from Johnston Atoll mentioned previously. After review of USNM and BPBM museum collections, we believe it is highly likely that the current known range of *C. matatakaro* reflects its true distribution, with Johnston Atoll remaining an uncertain but probable additional locality. The limited distribution of the new species within the central South Pacific and its status as sister to *Cirripectes vanderbilti* combine to strongly indicate a southern route-to-colonization for the Hawaiian endemic. Johnston Atoll has been shown to be a stepping-stone for biodiversity to enter the Hawaiian Islands (*Leray et al., 2010*; *Gaither et al., 2011*; *Andrews et al., 2014*; *Tenggardjaja, Bowen & Bernardi, 2014*), and thus constitutes a likely entry point to Hawaiʻi for the ancestor of *C. vanderbilti*. However, some studies have concluded that Hawaiian biodiversity may arrive directly from the Line Islands, bypassing Johnston Atoll (*Skillings, Bird & Toonen, 2010*; *Concepcion et al., 2016*). Additionally, *Randall, Lobel & Chave (1985)* remarked upon the conspicuous absence of a number of both wide-ranging and endemic Hawaiian fishes from Johnston Atoll. The new species described in this study occurs throughout the Northern Line Islands (where it makes up a majority of the specimens of *Cirripectes* sampled) but has yet to be collected from Johnston. In the absence of specimens of *C. matatakaro* from Johnston Atoll and given the limited genetic coverage we currently possess, both routes are possible, though both support the southern route hypothesis.

## CONCLUSIONS

The Hawaiian Archipelago is one of the most isolated island groups in the world and the origin of its marine species is an important question. Hawaiian biodiversity is thought to arrive either from the Western Pacific, via the Kuroshio Current, or from the south, via dispersal from Johnston Atoll or the Line Islands. The Scarface Blenny *Cirripectes vanderbilti*, which is endemic to the Hawaiian Islands and Johnston Atoll, was long thought to be closely related to the widespread *C. variolosus*. Through genetic and taxonomic analyses, we showed that the sister species to *C. vanderbilti* is a new species, *C. matatakaro*, that is known from the Line Islands south to the Marquesas, Pitcairn, Tuamotus, Gambier, and Austral Islands. Its limited distribution throughout islands to the south of Hawaiʻi and its status as sister to the Hawaiian endemic strongly indicates a southern route-to-colonization, although the lack of specimens from Johnston Atoll leaves the specific pathway an open question.

Our work, and other recent studies, shows that the genus *Cirripectes* contains more cryptic diversity than previously thought. These results highlight the importance of ongoing genetic and biodiversity inventories on coral reefs, particularly as these habitats are increasingly under threat. Investigation of often-overlooked groups such as cryptobenthic reef fishes may uncover interesting evolutionary patterns, as in the case of *Cirripectes*, where widespread taxa are found to comprise multiple cryptic lineages with adjunct geographic ranges, suggesting parapatry as well as allopatry as evolutionary mechanisms in fishes. Our work also showcases the value of natural history collections to taxonomic and biogeographic research. Despite low sample numbers and narrow geographic coverage for our phylogeographic/genetic analyses, we were able to use museum specimens to make inferences about species range sizes and historic routes to colonization. Combtooth blennies and other cryptobenthic fishes often utilize vulnerable, high-energy surge zone habitats and may be underrepresented in museum collections as these habitats are difficult and/or dangerous to sample. As mass coral bleaching events and habitat degradation increase worldwide, we risk extensive biodiversity loss before we are even aware of its existence.

## ACKNOWLEDGEMENTS

For field assistance we thank S Planes, E Delrieu-Trottin, P Sasal, J Mourier, M Veuille, R Galzin, T Lison de Loma, G Mou-Tham, M Gaither, M Iacchei, D Wagner, D Skillings, B Bowen, R Kosaki, S Karl, R Coleman, C Westbrook, D Kraft, and J Copus. For logistic support we thank David Pence and Jason Jones from the University of Hawaiʻi Dive Safety Office. For field and logistical support on Kiritimati Island, we thank Patrick Price, Taburuea Tomataake, Karakaua Marakia, and the Frigate Bird Resting Cross Villages at Tabwakea. We thank John E. Randall for taking the photograph used in Fig. 8 and for the donation of his photographic slides to the National Museum of Natural History. We thank David Rolla for the *in-situ* photograph in Fig. 10. For i-Kiribati language assistance, we thank Takuia Uakeia from University of the South Pacific, Tarawa and Taratau Kirata from Kiribati Fisheries Division, Kiritimati. We thank Janina Larenas for reminding us of

Dario Argento and his use of color, and for suggesting the common name. We gratefully acknowledge the technical support and advanced computing resources from the University of Hawaiʻi Information Technology Services –Cyberinfrastructure. We are grateful to T Frogier, P Mery and the Centre Plongée Marquises (Xavier (Pipapo) and Marie Curvat), for their field assistance in the Gambier, the Marquesas, and the Australs along with the crew of the *Claymore II*, *Braveheart* and the *Golden Shadow*. We thank the Ministère de l'Environnement de Polynésie, the Délégation à la Recherche Polynésie, the Mairie of Nuku-Hiva, and the people of the Marquesas Islands for their kind and generous support of the project as we traveled throughout the islands. We thank Jerry Finan, Erika Wilbur, Shirleen Smith, Kris Murphy, Diane Pitassy and Sandra Raredon of the Division of Fishes (National Museum of Natural History) for assistance in preparations for the French Polynesia trips and processing specimens, and Lee Weigt, Amy Driskell, Jeffrey Hunt and Kenneth Macdonald III and Meaghan Parker Forney of the Laboratories of Analytical Biology (Smithsonian Institution) for assistance in molecular analysis of samples. We thank the staff of the CRIOBE and particularly Yannick Chancerelle for logistical support in French Polynesia. We thank Brian Bowen for helpful comments and discussion regarding an initial draft of this paper. Finally, we thank Luiz Rocha and two anonymous reviewers for constructive comments and suggestions on an earlier version of this paper. Any findings, and conclusions or recommendations expressed in this material are those of the author(s) and do not necessarily reflect the views of the National Science Foundation. The views expressed herein are those of the author(s) and do not necessarily reflect the views of NOAA or any of its subagencies. This is contribution #JC-18-18 from the Hawaiʻi Sea Grant College Program, #1776 from the Hawaiʻi Institute of Marine Biology, and #10916 from the School of Ocean and Earth Science and Technology at University of Hawaiʻi.

### Funding

This material is based upon work supported by the National Science Foundation Graduate Research Fellowship Program under Grant No. DGE-1329626 and an internship provided through the Graduate Research Internship Program (GRIP) in cooperation with the Smithsonian Institution National Museum of Natural History Office of Internships & Fellowships. Mykle Hoban was also supported by a Graduate Traineeship funded by a grant/cooperative agreement from the National Oceanic and Atmospheric Administration, Project R/HE-31, which is sponsored by the University of Hawaiʻi Sea Grant College Program, SOEST, under Institutional Grant No. NA18OAR4170076 from NOAA Office of Sea Grant, Department of Commerce. UNIHI-SEAGRANT-JC-18-18. Mykle Hoban received research support from the Colonel Willys E. Lord & Sandina L. Lord Endowed Scholarship and The Explorers Club Exploration Fund Grant. DNA data production and field work were supported by the National Science Foundation (OCE-1558852 to B.W. Bowen), and University of Hawaii Sea Grant College Program. The work of Jeffrey T. Williams in French Polynesia was made possible by invitations from Rene Galzin and

Serge Planes to participate in expeditions that were financially supported by the French National Agency for Marine Protected Area in France ('Pakaihi I Te Moana' expedition), the ANR IMODEL and Contrat de Projet Etat-Territoire in French Polynesia and the French Ministry for Environment, Sustainable Development and Transport (MEDDTL) ('CORALSPOT' expeditions), and the Living Oceans Foundation ('Australs' expedition). Additional funding was provided by the IFRECOR in French Polynesia and the TOTAL Foundation. There was no additional external funding received for this study. The funders had no role in study design, data collection and analysis, decision to publish, or preparation of the manuscript.

**Grant Disclosures**

The following grant information was disclosed by the authors:
National Science Foundation Graduate Research Fellowship Program: DGE-1329626.
Graduate Research Internship Program GRIP.
Smithsonian Institution National Museum of Natural History Office of Internships & Fellowships.
National Oceanic and Atmospheric Administration: Project R/HE-31.
University of Hawai'i Sea Grant College Program, SOEST: NA18OAR4170076.
NOAA Office of Sea Grant, Department of Commerce: UNIHI-SEAGRANT-JC-18-18.
The Explorers Club Exploration Fund Grant.
National Science Foundation: OCE-1558852.
French National Agency for Marine Protected Area in France.
French Ministry for Environment, Sustainable Development and Transport (MEDDTL).
Living Oceans Foundation.
TOTAL Foundation.

**Competing Interests**

The authors declare there are no competing interests.

**Author Contributions**

- Mykle L. Hoban conceived and designed the experiments, performed the experiments, analyzed the data, prepared figures and/or tables, authored or reviewed drafts of the paper, and approved the final draft.
- Jeffrey T. Williams conceived and designed the experiments, performed the experiments, analyzed the data, authored or reviewed drafts of the paper, and approved the final draft.

**Animal Ethics**

The following information was supplied relating to ethical approvals (i.e., approving body and any reference numbers):

All vertebrates collected for this study were taken in accordance with University of Hawaii IACUC protocol 09-753-5, "Phylogeography and Evolution of Reef Fishes" (PI: Dr. Brian Bowen).

## Field Study Permissions

The following information was supplied relating to field study approvals (i.e., approving body and any reference numbers):

Specimens from the Northwest Hawaiian Islands and Johnston Atoll were collected under permit #PMNM-2018-031 from the Papahānaumokuākea Marine National Monument, issued to B. Bowen. Collections at Kiritimati were taken under permit #002/17 issued to B. Bowen by the Republic of Kiribati Environment and Conservation division. Specimens from French Polynesia were collected under permit number ''Permanent agreement, Délégation à la Recherche, French Polynesia'' All material from French Polynesia has been deposited in the Smithsonian National Museum of Natural History collections, and catalog numbers are given for each specimen examined in the text and supplemental information.

## DNA Deposition

The following information was supplied regarding the deposition of DNA sequences:

The COI sequences for specimens sequenced at HIMB are available at GenBank: MN649877 to MN650012.

The COI sequences for specimens from the NMNH collections are available at GenBank: MK566873 to MK566874, MK658095 and MK658122.

## Data Availability

Specimen data and R code used for analyses and figure generation is available at: mhoban. (2020, February 14). mhoban/cirripectes: manuscript resubmission data (Version v1.1). Zenodo. DOI 10.5281/zenodo.3666948.

A detailed list of specimens examined in this study with institutional catalog numbers are in a Supplementary File.

## New Species Registration

The following information was supplied regarding the registration of a newly described species:

Publication LSID: urn:lsid:zoobank.org:pub:E904BAB5-9C52-46A0-9B74-530C39F571DE
Cirripectes matatakaro LSID:
urn:lsid:zoobank.org:act:B9D062E5-6D3D-4218-B225-BE31147B025B.

## Supplemental Information

Supplemental information for this article can be found online at http://dx.doi.org/10.7717/peerj.8852#supplemental-information.

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
