# Peer review of "Cirripectes matatakaro, a new species of combtooth blenny from the Central Pacific, illuminates the origins of the Hawaiian fish fauna"

_PeerJ, doi:10.7717/peerj.8852_

## Round 0.1 · original submission · Minor Revisions

Thank you for submitting your manuscript to PeerJ. Based on the thorough and positive comments of three reviewers I invite you to resubmit after making minor revisions. All three reviewers agree that your data support the description of this new species. Most of their suggestions are on relatively minor points that would strengthen your paper. However, there are some more significant issues to address. First, reviewer 1 strongly recommends that at least one type specimen be deposited in an alternate collection prior to publication (this issue is also noted by reviewer 3). Second, reviewer 2 has concerns about the fossil data presented, especially the assignment of an otolith to genus and the time calibration of your phylogeny. Reviewer 3 notes similar concerns, suggesting that the phylogeny be removed all together and identifying problems with your use of molecular clock findings for this group.

Please be sure to respond to all reviewer comments in a response letter with your resubmission.

Reviewer 1 ·

Basic reporting

The paper is well written.

Experimental design

Solid.

Validity of the findings

Solid

Additional comments

It is always a pleasure to read a manuscript where your only role as a reviewer ends up being copyediting. I am completely convinced that this is a new species, and I agree with their morphological and molecular data.

I have a single substantive concern and maybe 56 copy-editing level concerns. My only real concern is that the type series is exclusively located in a single collection. This is frowned upon as all of the authors should know. This is particularly true given that the type material is exclusively near a country's capital. We don't have to look beyond Brazil to see that all material being located in a single collection is totally unacceptable (even if in a national collection). All journals, including PeerJ, should mandate that one of the type specimens is located in a second collection when the material is as common as seen in this study. It is difficult when someone describes a species including material from unaffiliated collections, but this study is submitted by the head collection manager of the relevant collection. I cannot understate how critical it is that the type series includes at least one specimen from a second collection; please transfer one specimen to a second institution.

Thanks again for the high-quality submission. Here are the copy-editing comments:

Throughout — the Oxford comma is not consistently used or not,

Title — should it say "the" Hawaiian fish fauna?

Second to last line of abstract, line 31 — combines should be combined

line 51 — sp. shouldn't be italicized.

line 54 — spp. shouldn't be italicized.

line 71 (most <100 mm SL) — should be followed by a comma

line 80 — should these numbers be written out?

line 83 — most references consider the common name to be a proper noun such that all words are capitalized.

line 117 — should this be fishes?

line 144 — comma should be removed

lines 149–152 — should this appear above the preceding paragraph because of the use of institutional codes in this preceding paragraph. Also why say NMNH here instead of USNM? Also, see line 155 where you given the National Museum as SECOND acronym. I get it, but how can the same place be abbreviated in two different ways. PICK ONE and use it consistently

lines 206–207, line 241, line 327, line 330 (x2), line 330–331, line 334, line 346 (x2), line 347, line 348 (x2), line 349 (x2), line 350 (x3), line 351, line 366, line 375, line 384 — normally ranges use an en-dash not a hyphen; please correct here and elsewhere

line 208, line 249, line 276, line 410, line 440, line 487, line 503, line 504 — it is usually frowned upon to use a scientific name as an adjective; consider a re-write

line 212 (first part) — here commas are used to separate the thousand levels, but they were not used in line 185 and 212 (second part); be consistent and follow the journal guidelines

line 302 – vessels are usually referred to as fishing vessels or research vessels

lines 320–323 — it is ALWAYS preferable to use a differential diagnosis where you provide the alternative states for the relevant taxa (congeners or congeners found in same location)

lines 339–340 — why use fin-rays or fin-spines here, but not on line 341?

line 354 — dorsal-fin spine (needs hyphen)

line 413 — why use a mononomial?

line 414 — needs a comma before the first and

line 416 — impractical is such a throw away; either delete or comment intelligently on its possibility or not using Sanger and high-throughput sequencing

line 445 — needs a comma before the and

line 451 — other species of what? Cirripectes, Blenniidae, Blennioidei, Percomorpha, Teleostei?

line 468–470 — needs a citation to support point

line 473 & line 483–486 — needs a comma before the but

line 504 — remove comma

line 526 — separate "and other recent studies" off by commas

Figure 2a lacks resolution in the version I can review

Figure 6 has filamentosus is misspelled

Figure 8 figure legend should say live not life

Again, I recommend acceptance after a specimen is transferred to a second institution and these copy-editing changes are made.

Reviewer 2 ·

Basic reporting

no comment

Experimental design

no comment

Validity of the findings

no comment

Additional comments

The authors of “Cirripectes matatakaro, a new species of combtooth blenny from the Central Pacific, illuminates the origins of Hawaiian fish fauna” present a well supported species description and add interesting phylogeographic comments. The authors do a nice job of setting up the interesting biogeography of Hawaii in the introduction. I am very impressed at the amount of work that went into this species description. There is a lot of research, knowledge and data gathering necessary to tease apart the relationship of a cryptic species, let alone within an already difficult group to identify (Cirripectes). A lot of care went into describing subtle differences among traits! I found the section on depth segregation rather interesting and wonder how common this is within closely related blenny species. I feel that the phylogenetic analyses might benefit from some updates (see species comments). One possible contentious point: the two fossils. Blenny fossils are exceedingly rare due to habitat preferences and other factors. In my opinion there has been no reliable time calibrated blenniid phylogeny. There are several relatively whole body fossils (see below), but most are known from just otoliths. I am not a fossil expert, but I find it hard to believe that an otolith can be placed into a genus. Especially a small, relatively simple shaped otolith. I feel that the authors should remove the dating aspect since it is barely discussed and possibly inaccurate. If the time calibrated phylogeny is included I suggest a sentence describing the fossils.

Bannikov, A. F. (1998). New blennioid fishes of the familes Blenniidae and Clinidae (Perciformes) from the
Miocene of the Caucasus and Moldova. Paleontological Journal, 32(4), 385–389.
Yabumoto, Y., & Uyeno, T. (2007). Tottoriblennius hiraoi, a new genus and species of Miocene blennioid fish
from Tottori Prefecture, Japan. Bulletin of the National Museum of Nature and Science. Series C, Geology &
Paleontology, 33, 81–87.
L60-65: This section needs to be reworded and updated. I think Brandl recognized more families as
cryptobenthic… currently reads like there are 4. Use of “They” is a bit awkward. To avoid confusion refer to
blenniids as Combtooth Blennies and thereafter blennies.
L69: Perciformes is confusing and inaccurate. Please choose a different clade. Ovalentaria? See Betancur-R et
al 2017 or Near et al 2013
L72: Citation for Cirripectes as herbivores?
L77: Canines are on Premaxillary and dentary bones
L78-81: Williamsichthys is a named clade for Cirripectes sister genera
L104-114: I feel this can be moved to acknowledgments. NMNH abbreviation is used as before it is spelled out
L135: “In addition” used consecutively. Change one
L136: define nuchal flap
L146: BPBM abbreviations is used as before it is spelled out
L172: More accurately Exallias + Ophioblennius + Scartichthys is the sister clade to Cirripectes.
L176-187: was this data partitioned by codon? If not it probably should be, if so seems unlikely that
jmodeltest2 would select the same model for codon position 1,2,3 - especially a complex model. A poorly fit
model may be contributing to the number of poorly supported nodes in Figure 6.
L195: citation for pegas
L197-221: How many sequences (individuals) were used to represent each tip in BEAST analysis? Is it 1? IS this
because BEAST requires species designation? Seems like authors throwing a lot of data to the side, especially
since topology in Fig 6 is slightly different from Fig 7. Not sure how node at C. stigmaticus, variolosus,
casatneus in Fig. 6 is so highly supported. Is there a zero length branch at that node in Fig. 6?
L 320 – phylogenetic?
L334 (8): clarify if color in life?
L465-684: Critical point!
L580: Not sure if this citation is correct. I was unaware of this fossil and found a different citation… but not
sure which is correct.
L585, L663, L669: some parts of citation are in all caps
Figure 1: I think the figure would benefit from having the phylogeny (even cartoon version w/ fewer tips)
color/shape. Would help readers not as familiar with localities see where individuals/species were found.
Figure 2A: Photo color looks odd, but may be just a low resolution for this review version.
Figure 5B: It might help to label which species is lineage A and B

·

Basic reporting

This is a very well-written paper describing a new species of blenny from the central/south Pacific and placing it in an evolutionary context. Even though I am suggesting several changes in the comments to author box, these are all minor, with the possible exception of the removal of the phylogenetic analysis and correction of the molecular clock.

Experimental design

Research questions are clear and methods are appropriate, except for the phylogenetic analysis, which I think should be removed. In general, the data strongly supports the conclusions.

Validity of the findings

The authors make a strong case for recognition of Cirripectes matatakaro as a new species.

Additional comments

Minor suggestions:

Sentence in lines 35-36 (and line 514) suggests that Hawaii is the most isolated island in the world. I would change the sentence from "is the most isolated island chain in the world" to "is one of the most isolated island groups in the world". I know there are no other more isolated CHAINS but the readers might not know the difference between a chain and and archipelago and assume that you are saying that Hawaii is the most isolated archipelago in the world, which is not the case.

Line 37: Replace "Hawai'i hosts a highly endemic marine fauna, with the proportion of endemism among marine fishes having been recorded at 25%" with "a high rate endemism is found among the marine fauna of Hawai'i, with the proportion of endemics among fishes recorded at 25%".

Consistency: "Hawai'i" or "Hawaii". I don't have an opinion to which way to do it, but it's spelled as "Hawaii" in some places (line 25, 32, 552 are the ones I found), maybe pick one and use find and replace all.

Line 67: spell out eight (and every numeral that is not a count under 10).

On line 68, "The genus Cirripectes" should be the start of a new paragraph.

Lines 73, 74 need a citations for Cirripectes biology.

Replace "sister" with "closely related" on line 78. There should be only one sister genus, but may be several closely related. Same for line 182, Exallias is either "the" sister genus or "a" closely related genus.

Lines 154-173 could be reduced. If you are citing papers where we got the methods from, you don't need to specify the methods (i.e.: no need to write out primers and PCR conditions if they follow methods in Baldwin et al 2009, etc).

Lines 201-203, the 2% per million year rate presented by Bowen et al and Reece et al is for cytochrome B, not COI. The rate for COI is different, you have to use appropriate citations here and the appropriate rate for your marker. And the Bowen et al paper was published in 2001, not 2009. That's wrong on both the text and literature cited (which contain many other errors that I hope are corrected during the editorial process).

Lines 223-233, is this really necessary? I've never seen this statement in any other paper, including online journals.

Figure 5: please split the lineages by species (not lineage A and B). If they are split in the tree on Figure 6, they should also be split in the haplotype network by species (not lineage A and B under Cirripectes "variolosus").

I would delete lines 266-292 and analyses related to them. This adds very little to the manuscript and support for it is uncertain. First, the phylogenetic analysis are almost entirely absent from the discussion and conclusions, and second you can't really trust a phylogeny based on just one mitochondrial DNA marker, and this whole section does not contribute to the main subject of the paper (the new species description).

Paratypes: please deposit one or two paratypes in other institutions. Having them all in one collection (USNM) decreases access and increases risk.

Diagnosis: "based upon molecular phylogenic position" is not really a diagnostic character. Actually, I would delete lines 320-323 and leave the good diagnosis from lines 325-336.

First part of line 339 should be in methods.

Description: pectoral fin ray counts?

Color in life: Figure 8 shows a big difference between male and female colors, this should be discussed here.

---

## Round 0.2 · accepted · Accept

Thank you for resubmitting your manuscript and your work to address the comments of the reviewers. All three original reviewers are satisfied that you have addressed their suggestions and I am happy to now accept your paper for publication in PeerJ. Reviewer 2 has two minor comments that you could address in the final draft.

You will be given the option to make the reviews of your manuscript available to readers. Please consider doing so as this review record can be a great resource for readers of your paper and contributes to more transparent science.

Thank you for choosing PeerJ as a venue for publishing your work.

Reviewer 1 ·

Basic reporting

I just read through the changes, and I have no problems with the paper now.

Experimental design

I just read through the changes, and I have no problems with the paper now.

Validity of the findings

I just read through the changes, and I have no problems with the paper now.

Additional comments

I just read through the changes, and I have no problems with the paper now.

Reviewer 2 ·

Basic reporting

no comment

Experimental design

no comment

Validity of the findings

no comment

Additional comments

I really enjoyed reviewing this paper and feel that it is a nice contribution to our understanding of Hawaiian fish fauna. I only have two comments for this round of reviews:
line 75 – herbivorous is misleading, unless defined as including detritus
Line 112 – define CRIOBE

Thanks and best wishes.

·

Basic reporting

This is a greatly improved version, the authors addressed all of my suggestions and I think this is ready to go.

Experimental design

Experimental design and methods are appropriate.

Validity of the findings

The new species is valid, and this is an important finding highlighting how little we know about even this widespread species (that was really two).

Additional comments

I think this is ready to be published after final checks. I agree with the authors that differential diagnosis should be in a comparisons section.